# Tumor Treatment by Nano-Photodynamic Agents Embedded in Immune Cell Membrane-Derived Vesicles

**DOI:** 10.3390/pharmaceutics17040481

**Published:** 2025-04-07

**Authors:** Zhaoyang He, Yunpeng Huang, Yu Wen, Yufeng Zou, Kai Nie, Zhongtao Liu, Xiong Li, Heng Zou, Yongxiang Wang

**Affiliations:** Department of General Surgery, Second Xiangya Hospital, Central South University, Changsha 410011, China; 238211117@csu.edu.cn (Z.H.); 218202108@csu.edu.cn (Y.H.); wenyu2861@csu.edu.cn (Y.W.); 2204170417@csu.edu.cn (Y.Z.); 2204170413@csu.edu.cn (K.N.); 158202088@csu.edu.cn (Z.L.); lixionghn@csu.edu.cn (X.L.)

**Keywords:** photodynamic therapy, nano-photodynamic agents, immune cell membrane, immunotherapy, chemotherapy

## Abstract

Non-invasive phototherapy includes modalities such as photodynamic therapy (PDT) and photothermal therapy (PTT). When combined with tumor immunotherapy, these therapeutic approaches have demonstrated significant efficacy in treating advanced malignancies, thus attracting considerable attention from the scientific community. However, the progress of these therapies is hindered by inherent limitations and potential adverse effects. Recent findings indicate that certain therapeutic strategies, including phototherapy, can induce immunogenic cell death (ICD), thereby opening new avenues for the integration of phototherapy with tumor immunotherapy. Currently, the development of biofilm nanomaterial-encapsulated drug delivery systems has reached a mature stage. Immune cell membrane-encapsulated nano-photosensitizers hold great promise, as they can enhance the tumor immune microenvironment. Based on bioengineering technology, immune cell membranes can be designed according to the tumor immune microenvironment, thereby enhancing the targeting and immune properties of nano-photosensitizers. Additionally, the space provided by the immune cell membrane allows for the co-encapsulation of immunotherapeutic agents and chemotherapy drugs, achieving a synergistic therapeutic effect. At the same time, the timing of photodynamic therapy (PDT) can be precisely controlled to regulate the action timing of both immunotherapeutic and chemotherapy drugs. This article summarizes and analyzes current research based on the aforementioned advancements.

## 1. Introduction

Conventional treatment modalities for anti-tumor therapy include surgical resection, radiotherapy, chemotherapy, and molecularly targeted therapy. When it comes to treating early-stage malignancies, these treatments work incredibly well, yet they are still useless when it comes to treating advanced cancers [1]. Photodynamic therapy (PDT), as a minimally invasive treatment approach with years of clinical validation, has evolved from an experimental method to a clinically recognized adjuvant therapy, potentially offering new hope to patients with advanced cancer. The modern development of PDT began in 1975 when the team led by American scientist Thomas Dougherty systematically clarified the photosensitization mechanism of hematoporphyrin derivative (HpD) and successfully treated transplanted tumors in mice for the first time through PDT [2]. PDT is used to destroy tumor tissues by using photosensitizers (PSs) to selectively accumulate in tumor tissues and excite them under specific wavelengths of visible light irradiation to produce a large amount of reactive oxygen species (ROS) in tumor cells, killing tumor cells and inhibiting their growth [3,4,5]. With the advancements in photochemistry and nanotechnology, photosensitizers have evolved through three generations. First-generation photosensitizers, primarily represented by HpD, exhibit significant limitations such as pronounced skin phototoxicity and inadequate tissue penetration depth [6]. Second-generation photosensitizers, including 5-aminolevulinic acid (5-ALA) and temoporfin (mTHPC), have been chemically modified to optimize absorption wavelengths and metabolic stability, thereby enhancing their targeting efficacy and safety profile [7]. Third-generation photosensitizers leverage nanocarriers (e.g., liposomes, polymer micelles) or conjugation with targeting molecules (e.g., antibodies, peptides) to achieve tumor-specific accumulation and modulation of the immune microenvironment [8,9]. PDT is characterized by minimal side effects, low likelihood of drug resistance, and low systemic toxicity, making it a preferred choice for cancer treatment. In recent years, PDT has gained widespread application as a minimally invasive adjuvant therapy for tumors, demonstrating promising potential in both in vivo and in vitro studies [10].

The oxygen-consuming nature of PDT is one of its disadvantages [11], though, as most PDTs depend heavily on oxygen concentrations. In particular, the tumor environment itself has a hypoxic environment [12]. Hypoxic processes within the tumor microenvironment arise from the rapid proliferation of cancer cells, resulting in a significant mismatch between oxygen demand and supply, alongside notable metabolic alterations [13]. This situation severely impedes the production of ROS in PDT, thereby diminishing the ROS-induced endoplasmic reticulum (ER) stress and immunogenic cell death (ICD) effects [14,15]. This reduction weakens the effectiveness of oxygen-dependent PDT, while the oxygen consumption associated with PDT can exacerbate tumor hypoxia, fostering a detrimental feedback loop [16]. Moreover, the short excitation light wavelength characteristic of many PDT approaches results in poor tissue penetration and reduced targeting efficiency for tumor tissues. Consequently, some photosensitizers may accumulate in normal tissues, posing a risk of local phototoxicity and inducing photosensitivity reactions in the skin or mucous membranes upon light exposure [17]. Due to the limited depth of penetration, PDT alone is mostly only suitable for the treatment of superficial tumors.

Tumor immunotherapy can prevent tumor recurrence and prolong patient survival by activating or enhancing the host’s immune system to attack tumor cells and elicit specific anti-tumor immune responses and long immunogenic memory responses [18,19]. To date, numerous immune-based therapies have received approval for cancer treatment [20]. However, low patient response rates due to the immunosuppressive microenvironment, as well as the resulting adverse effects, remain major barriers to the use of widespread immunotherapy. Therefore, there is an urgent need to find a combined treatment modality that can bring good synergy and reduce the corresponding adverse effects. In the past, the concept of ICD has emerged, which is a form of cell death that stimulates both innate and adaptive immune responses and, as a result, long-term immune memory [21,22]. Various treatments have been found to induce ICD, with PDT standing out as an effective method for eliciting ICD [23]. Following ICD induction, damage-associated molecular patterns (DAMPs) like calreticulin (CRT), adenosine triphosphate (ATP), heat shock proteins (HSP70 and HSP90), and high mobility group box 1 (HMGB1) are released, exposing tumor antigens to antigen-presenting cells (APCs), thereby enhancing dendritic cell (DC) maturation, T-cell activation, and cytotoxic T lymphocyte (CTL) infiltration to activate the host immune system against cancer, instigating an anti-tumor immune response [19,24,25].

The ability of cancer therapy to trigger ICD is crucial clinically, as it engenders an anti-cancer immune response essential for therapeutic effectiveness and the maintenance of long-term anti-cancer immunity [26,27]. PDT induces ICD to promote the anti-tumor immune response, which builds a bridge between PDT and cancer immunotherapy [28]. Dendritic cell (DC)-based cancer vaccines represent a personalized immunotherapy approach that harnesses the patient’s own immune system. The core of this approach is to activate and enhance the antigen-presenting function of DCs to induce specific T-cell immune responses and thereby kill tumor cells [29]. Photodynamic therapy-induced ICD releases a large amount of tumor antigens, enhancing the antigen loading efficiency of DC vaccines and facilitating the development of cancer immunotherapy based on DC vaccines [30,31]. Combining PDT with immune adjuvants not only boosts PDT’s efficacy but also enhances immune cell activity markedly. Additionally, linking PDT with monoclonal antibodies (MAbs) can enhance photosensitizer specificity and reduce side effects. Incorporating checkpoint inhibitors (PD-L1/PD-1) with PDT induces regression of both light-irradiated primary tumors and distant, unirradiated tumors by triggering a potent tumor-specific immune response [32]. Therefore, the combination of PDT and immunotherapy presents promising prospects for future cancer treatment. However, key challenges such as the dependence on oxygen in the hypoxic tumor microenvironment, insufficient tissue penetration, and tumor targeting of therapeutic drugs still exist. These limitations necessitate further optimization of synergistic mechanisms and drug delivery strategies to fully realize the clinical potential of this combined approach.

In order to alleviate these problems in PDT, many new photosensitizers and combination therapies have been developed to improve the therapeutic effect of PDT. Third-generation photosensitizers have significantly improved tumor cell targeting while reducing associated side effects [32,33]. Nanomedicine-based delivery systems present promising prospects by addressing tumor hypoxia through various self-oxygen supply strategies [33,34]. Integration with chemotherapy and immunotherapy enables the combined application of multiple anti-tumor treatments, leading to synergistic effects [35,36]. Among them, PDT combined with immunotherapy can effectively suppress primary and distant tumors, which may be beneficial for the treatment of metastatic tumors [19].

In recent years, there has been an increasing number of studies on nanoparticle delivery systems based on biomembrane encapsulation. These biological membranes include red blood cell membranes (RBCMs) [37,38], cancer cell membranes (CM) [39], and immune cell membranes, etc. Red blood cell membrane nanomaterials have attracted the attention of many researchers because of their good biocompatibility and degradability, which can achieve long-term circulation in the body by endowing the delivered drug with the “stealth” property, so as to avoid the drug being cleared by the immune system before it takes effect, and also has a certain targeting [40,41,42]. Cancer cell membrane nanomaterials have attracted the attention of researchers for their good tumor targeting due to tumor homing effect and also have excellent immune escape ability from macrophages, which can achieve more precise drug delivery and improve treatment outcomes [43,44]. Compared with red blood cell membrane and cancer cell membrane nanomaterials, immune cell membrane nanomaterials retain a large number of immune cell surface proteins, which can activate or inhibit different immune cells, thereby regulating the proportion of different immune cells in the body. This plays a great role in promoting the remodeling of the tumor immune microenvironment and improving the environment for tumor immunotherapy, and has more advantages in promoting tumor immunotherapy. These materials exhibit precise targeting of tumor cells and the inflammatory environment, evading immune clearance and accruing in tumor tissues, suggesting a promising avenue for enhancing the synergistic effects of PDT and tumor immunotherapy. In recent years, research on nanomedicine delivery platforms embedding nano-photosensitizers into immune cell membranes has gradually increased (Figure 1). However, there is still a lack of relevant reviews to introduce the nanoparticle delivery system of immune cell membranes, and this review will introduce the mechanism of action and its impact on tumor treatment of the combination of various immune cell membrane nanoparticle-encapsulated photosensitizers and other drugs, and put forward possible directions for further research.

## 2. Advantages and Limitations of Nano-Photosensitizers

As an emerging class of nanomaterials, nano-photosensitizers have garnered significant attention in the medical and biological fields in recent years. Their unique photoexcitation properties enable them to generate targeted chemical effects under specific wavelengths of light, making them applicable to various therapeutic modalities such as PDT. Researchers have developed diverse types of nano-photosensitizers, including phosphorescent materials, carbon dots (CDs), and metal–organic frameworks (MOFs), to enhance both their photosensitivity and biocompatibility. In the field of photodynamic therapy, the application of nano-photosensitizers has shown significant potential. Phosphorescent materials have attracted much attention due to their potential as photosensitizers in PDT. Through energy transfer from the triplet state of phosphors to the ground state of molecular oxygen, highly cytotoxic singlet oxygen (^1^O_2_) can be efficiently generated, thus endowing these NPs with excellent performance in PDT [45]. Zhou et al. designed and synthesized water-soluble and multifunctional phosphorescent conjugated polymer dots (Pdots) [46]. Due to the incorporation of an oxygen-sensitive phosphorescent Pt(II) porphyrin complex, this Pdot exhibits a singlet oxygen quantum yield as high as 0.80, thereby demonstrating outstanding cancer cell-killing ability. MOFs have been used as novel nano-photosensitizer carriers, achieving breakthroughs in the field of photodynamic therapy (PDT). The first reported study on PDT based on porphyrin MOFs utilized the DBP-UiO nanosheet system [47], which had a porphyrin loading capacity of up to 77 wt.%, significantly enhancing the singlet oxygen (^1^O_2_) generation efficiency to more than twice that of free 5,15-di(p-benzoato)porphyrin (DBP) and significantly improving the PDT efficacy against human head and neck squamous cell carcinoma (SQ20B) cells. Jihye Park et al. integrated the photochromic molecule BPDTE (1,2-bis(2-methyl-5-(pyridin-4-yl)thiophen-3-yl)cyclopent-1-ene) into the MOFs framework to construct SO-PCNSO-PCN MOFs [48]. SO-PCN MOFs achieved reversible control of singlet oxygen generation through a competitive energy transfer pathway under specific wavelengths of light, demonstrating great potential in controllable PDT. In response to the hypoxic microenvironment of tumors, Lan’s team innovatively designed Fe-TBP nano-MOFs [49], where the Fe_3_O nodes catalyzed the decomposition of intratumoral H_2_O_2_ through the Fenton reaction to continuously generate O_2_. Under hypoxic conditions, Fe-TBP exhibited comparable PDT efficacy with an IC50 of 3.10 ± 1.66 μM, while Hf-TBP and H4TBP were completely ineffective, with IC50 values far greater than 50 μM. Compared with the Hf-TBP system, Fe-TBP demonstrated the best PDT effect under both normoxic and hypoxic conditions (cell viability decreased by approximately 15% and 65%, respectively). Studies have shown that certain nano-photosensitizers can efficiently generate superoxide radicals and singlet oxygen in water, which is crucial for the killing effect on tumor cells [50]. CDs as a new type of fluorescent nano-photosensitizer and its carrier, possess the characteristics of simple synthesis process, low biological toxicity, and easy surface functionalization modification [51]. Combined with excellent water solubility and high singlet oxygen quantum yield, they can not only serve as efficient PS but also as a drug delivery platform. Research has confirmed that CDs can effectively improve the shortcomings of traditional photosensitizers such as insufficient water solubility and low bioavailability, and significantly enhance cellular uptake efficiency by optimizing drug diffusion kinetics [52]. For instance, Wen et al. synthesized near-infrared emitting CDs using magnesium-free chlorophyll as the precursor by microwave method [53], which has both a singlet oxygen quantum yield (0.62) and tumor-targeting imaging function, and can be used as a low-toxicity and highly efficient PS for cancer treatment. To address the water solubility and pharmacokinetic defects of the commonly used PS chlorin e6 in clinical practice, Songeun Beack et al. chemically conjugated CDs, HA, and Ce6 to solve these problems [54]. Additionally, carbon dots exhibit excellent bioimaging capabilities and PDT efficacy, effectively generating ROS even in hypoxic conditions, offering innovative strategies for cancer treatment [55].

Although nano-photosensitizers have shown good effects in laboratory research, they still face many challenges in clinical applications. First, the biocompatibility and safety of nanomaterials remain key concerns for clinical application. Second, environmental factors significantly impact the stability of nanomaterials, including temperature, pH value, ionic strength, and light exposure. These factors not only influence the physical and chemical properties of nanomaterials but can also lead to degradation or aggregation in specific environments. Elevated temperatures may accelerate nanoparticle aggregation, reducing dispersion and stability; changes in pH can alter surface charge, affecting behavior in liquids [56]. Nanomaterials synthesized from bio-based materials are considered a sustainable development direction due to their lower toxicity and environmental impact [57]. Furthermore, optimizing synthesis methods has been shown to significantly improve environmental compatibility and biocompatibility, promoting broader applications in biomedical engineering and environmental remediation [58].

## 3. The Comprehensive Advantages of Immune Cell Membrane-Embedded Nanomaterials

In recent years, in the construction of nanomedicine delivery platforms, cell membranes, as a kind of natural functional material, have gradually become a research hotspot. Owing to their unique biological properties, different types of cell membranes have garnered significant attention for their distinct functional capabilities in drug delivery, cancer therapy, immune modulation, and other biomedical applications (Figure 2). Notably, immune cell membranes stand out due to their superior targeting ability and potent immune regulatory functions, demonstrating substantial potential for innovative therapeutic strategies.

Immune cell membranes retain a large amount of natural membrane information, enabling them to provide targeted functions for specific antigens and diseased cells in drug delivery and cancer therapy, while also participating in immune regulation and immune responses [59]. In contrast, red blood cell membranes exhibit high biocompatibility and immune evasion capabilities but lack active targeting ability [60,61]. Cancer cell membranes possess homologous targeting capacity towards cancer tissues [62], but they pose immunogenicity risks, including potential autoimmune reactions, the presence of molecules that promote cancer progression, and uncertain pathogenicity when used as a cancer vaccine [63,64]. Nanomedicines encapsulated by bacterial membranes have inherent immune activation properties [65], but they may induce excessive inflammatory responses. Platelet membranes demonstrate damage-targeting capability [66] and are suitable for targeting vascular injuries and tumor microenvironments [67]. In terms of functionality, immune cell membranes exhibit significant advantages over other biological membranes. Leveraging the specific binding of immune cell receptors to antigens [68] and their targeting ability for inflammatory sites and tumor microenvironments, they can precisely identify and target diseased sites, offering the highest targeting performance. In contrast, red blood cell membranes have lower targeting ability, while cancer cell membranes have homologous targeting ability and strong targeting ability for cancers of the same origin. Moreover, immune cell membranes possess potent immune regulatory capabilities, with different types of immune cell membranes exhibiting unique immune modulation effects [69,70], providing effective immune enhancement effects for cancer treatment. Red blood cell membranes and cancer cell membranes have weaker immune regulatory capabilities, while bacterial membranes, although having immune activation ability, may trigger excessive inflammatory responses [71,72], necessitating further evaluation of their biosafety. In comparison, the inflammatory response elicited by immune cell membranes is controllable, thereby avoiding tissue damage caused by excessive inflammation.

In the application of biomembranes, immune cell membranes exhibit superior targeting capabilities, making them ideal carriers for precise drug delivery and high-sensitivity diagnostic imaging at tumor and inflammatory sites [73]. In contrast, red blood cell membranes possess limited targeting ability and are primarily utilized for long-circulating drug delivery and imaging applications [74]. While cancer cell membranes offer specific advantages in tumor targeting, cancer vaccine development, and tumor-specific imaging [75], their applicability is predominantly limited to homologous tumors. Furthermore, their biological safety necessitates thorough evaluation. In cancer therapy, immune cell membranes can be integrated with immunotherapy to enhance the body’s anti-tumor immune response [59,76], thereby improving the efficacy of cancer treatment. Furthermore, immune cell membranes can restore immune system balance through immune regulation, offering potential applications in autoimmune diseases and inflammatory disorders. Red blood cell membranes, on the other hand, primarily provide immune evasion properties [77].

Ultrasound technology facilitates the embedding of nano-photosensitizers into immune cell membranes. Research indicates that ultrasound not only enhances nanoparticle loading efficiency but also improves stability and biocompatibility [78], leading to better uniformity and biological activity of the final product [79]. Additionally, self-assembly methods can be employed to embed nano-photosensitizers within immune cell membranes. The self-assembly approach effectively generates more stable membrane structures, enabling nanoparticles to self-assemble on or within the cell membrane, forming robust nanocomposites. Studies have shown that self-assembly increases nanoparticle loading capacity and enhances adaptability and stability in biological environments [80]. Moreover, this method is highly tunable, allowing optimization of nanomaterial properties by adjusting environmental conditions such as pH and ionic strength to meet diverse application requirements [81]. Zeta potential is an important parameter reflecting the surface charge characteristics of nanoparticles, which directly affects the stability of colloids and the interaction between particles. In the field of drug delivery, zeta potential data is often used to evaluate the stability of colloids. According to literature reports, nanoparticle dispersion systems with zeta potentials of ±0–10 mV, ±10–20 mV, ±20–30 mV, and ±30 mV are generally classified as highly unstable, relatively stable, moderately stable, and highly stable, respectively [82]. Generally, it is believed that when the absolute value of zeta potential is larger, the stability of nanodrugs is stronger; however, zeta potential cannot completely predict the behavior of nanoparticles in blood [83]. In the research we are concerned about, after the immune cell membrane wraps the nanodrugs containing photosensitizers, the average particle size increases, and the absolute value of zeta potential of most samples significantly rises and is close to the zeta potential of the immune cell membrane itself. Their experimental results all indicate that the nanodrugs wrapped by the immune cell membrane exhibit excellent stability.

## 4. Enhancement of the Stability and Targeting Efficacy of Nano-Photosensitizers via Immune Cell Membranes

Adhesion molecules are widely expressed on the surface of immune cells and can assist in the enrichment of nano-photosensitizers encapsulated in immune cell membranes at tumor sites through the interaction with adhesion molecule ligands ICAM-1 on vascular endothelium [84,85,86,87]. The interaction between receptors on the surface of immune cells and antigens on the surface of tumor cells is the basis for immune cells to recognize tumors (Figure 3). Immune cell membrane-embedded nano-photosensitizers can recognize tumors by taking advantage of this property, and since there is no subsequent signal transduction, the influence of receptors in the subsequent process does not need to be considered [88,89]. For these reasons, embedding nano-photosensitizers in immune cell membranes can endow them with the ability to target tumors, thus facilitating their enrichment at tumor sites [59]. Immune cells are mainly divided into innate immune cells (macrophages, neutrophils, NK cells, etc.) and adaptive immune cells (T cells), and the tumor recognition ligands on the cell membranes of different immune cells may vary, corresponding to different tumor immune processes. In addition, using immune cell membranes to encapsulate nanomaterials can avoid clearance in the liver due to the biocompatibility of immune cell membranes and prevent the diffusion of nano-photosensitizers in the body, reducing their organ toxicity (Figure 4).

### 4.1. Macrophage Membrane

Macrophages typically possess anti-tumor activity and can inhibit tumor growth by phagocytosing tumor cells and secreting pro-inflammatory factors [90,91]. The macrophage membrane, through specific proteins, can not only recognize tumor cells but also activate tumor immunity [92,93]. The main proteins expressed on the macrophage membrane include phagocytic receptors and cell adhesion molecules, which play a key role in tumor antigen recognition [66,67]. Furthermore, the interaction between CD47 on tumor cells and SIRP-α on macrophage surfaces constitutes an immunosuppressive pathway. However, materials encapsulated within immune cell membranes do not activate subsequent immunosuppressive signaling pathways. Consequently, macrophage membrane-encapsulated immune nano-photosensitizers can recognize tumor cells via SIRP-α without inhibiting subsequent immune responses [94]. Additionally, the Fc receptor (FcγR) on macrophage surfaces binds to IgG or IgM antibodies in immune complexes, enabling the macrophage membrane to facilitate tumor recognition through antibody-mediated mechanisms [95].

By leveraging the unique properties of macrophage membranes, cell membrane-camouflaged nanoparticle technology achieves more effective drug delivery, thereby enhancing cancer treatment efficacy [96]. Researchers have ingeniously used macrophage membranes to coat nanoparticles, significantly increasing their circulation time in the bloodstream and endowing them with superior tumor-targeting capabilities, enabling efficient accumulation in tumor tissues [97,98].

Nanomaterials embedded within macrophage membranes can prolong retention in the bloodstream, exhibit excellent biocompatibility, and have negligible systemic side effects. Gao et al. constructed macrophage membrane-anchored nano-photosensitizers (CMA-nPS) by fusing azide-modified macrophage membranes with vesicular stomatitis virus glycoprotein (VSVG)-modified NIH3T3 cell membranes and embedding dibenzocyclooctyne (DBCO) [99]. This approach significantly reduced the clearance rate of CMA-nPS and prolonged its retention in the blood. Xie et al. self-assembled a near-infrared photosensitizer chlorin 6-C15-ethyl ester (HB) and an indoleamine-(2,3)-dioxygenase (IDO) pathway inhibitor NLG8189 into small molecules and camouflaged them with macrophage membranes to build the nano-drug HB-NLG8189@MPCM [100]. They also demonstrated that the macrophage membranes reduced non-specific clearance and prolonged blood retention time. Besides reducing clearance in the blood and thus extending the circulation time of nanoparticles in the blood, nanomaterials embedded in macrophage membranes also have a decreased clearance efficiency by target cells after being absorbed, thereby increasing the accumulation concentration and prolonging the action time. Sun et al. developed a nano-therapeutic agent (AuND-TPP-ICG@MCM) based on gold nanodendrites (AuND) [101]. The macrophage membrane coating not only extended the circulation time but also enhanced drug accumulation in cancer cells, particularly within mitochondria. Hu et al. engineered a stimulus-responsive nanoparticle [(C/I)BP@B-A(D)&M1m] loaded with doxorubicin (DOX) and an indoleamine 2,3-dioxygenase 1 inhibitor (IDO1) [102]. The nanoparticles coated with macrophage membranes showed excellent stability and effectively delayed the clearance of the mononuclear phagocytic system.

Drug targeting is a critical issue in disease treatment. In addition to enhancing the stability of nanomaterials, macrophage membranes improve their targeting ability, thereby enhancing therapeutic efficacy. Wang et al. utilized M1 macrophage-derived cell membrane vesicles, TAPP, and Cu^2+^ to construct the M-Cu-T drug delivery system [103]. In vivo experiments confirmed that the macrophage membrane enhanced M-Cu-T’s tumor retention ability. Chen et al. synthesized a photosensitizer based on rare earth upconversion nanoparticles (UCNP) conjugated with Rose Bengal (NPR) and coated it with tumor-associated macrophage membranes (TAMM) to construct NPR@TAMM [104]. The macrophage membrane not only improved retention at the tumor site but also enhanced binding and uptake by tumor cells. Meng et al. constructed a nano-core with immune adjuvant and aggregation-induced emission properties via ultrasonic treatment and coated it with macrophage membranes to produce M@PFC [105]. The macrophage membrane coating enabled the nano-drug to evade immune system clearance, prolong circulation time in the body, and enhance accumulation in tumor tissues by interacting with vascular cell adhesion molecule 1 (VCAM-1) on cancer cells. The M@BS-QE NPs constructed by Zhao et al. were recruited to the tumor surface through the CCL-2-dependent mechanism brought by the macrophage membrane, thereby endowing the M@BS-QE NPs with excellent tumor-targeting ability [106]. Biological barriers pose significant challenges to drug therapy and reduce targeting efficiency. Due to their good tissue compatibility and biocompatibility, macrophages have been utilized by Liu et al. to encapsulate nano-platinum in liposomes, load the hydrophobic clinical photosensitizer verteporfin in the lipid bilayer, and coat it with RAW264.7 macrophage cell membranes to form nano-Pt/VP@MLipo [107]. The results showed that the macrophage membrane enabled nano-platinum to penetrate the tumor barrier more easily, thereby exerting cytotoxic effects in deeper tumor tissues.

The numerous characteristics of macrophage membranes can help encapsulate drugs and improve their clinical properties, achieving the goal of precision medicine. Yu et al. constructed a CuS/carbon dot nanocomposite (CuSCD) and coated it with macrophage membranes hybridized with T7 peptide to form CuSCDB@MMT7 [108]. After being coated with macrophage membranes, the system exhibited sustained encapsulation efficiency and excellent stability under physiological conditions and could significantly control the release of chemotherapeutic drugs. Wang et al. constructed liposomes containing NONOate, IR780, and perfluorocarbon, co-extruding them with macrophage membranes and IR780-NO-PFH-Lip to form a biomimetic IR780-NO-PFH-Lip@M [109]. The macrophage membrane-coated nano-drug delivery platform significantly enhanced accumulation in xenograft tumors, prolonged circulation time in the blood, and achieved precise treatment of tumor tissues.

In summary, the nano-drug delivery system coated with macrophage membranes has exhibited remarkable advantages and potential in tumor therapy. Owing to its protein profile being highly analogous to that of native macrophage membranes, this system can effectively escape immune surveillance and significantly extend circulation time in vivo. Moreover, the macrophage membranes possess intrinsic targeting properties toward the tumor microenvironment and can interact with tumor-specific adhesion molecules, thereby conferring superior tumor-targeting capabilities to the nano-drugs. The synergistic effects of immune evasion and active targeting enable drugs to accumulate more efficiently within tumor tissues while minimizing non-specific distribution, thus enhancing both biosafety and therapeutic efficacy. With the continuous advancement of technology and the deepening of research, such nano-drugs are expected to play a more significant role in tumor treatment.

### 4.2. Neutrophil Membrane

Neutrophils (NEs), as essential components of the immune system, not only initiate immune responses but also play a pivotal role in linking inflammation and cancer. Leveraging their inherent tumor tropism, NEs can deliver therapeutic agents precisely to tumor sites [110,111]. Importantly, NEs can release payloads in response to inflammatory signals, making them ideal carriers for enhancing drug delivery within tumors [112,113]. Neutrophils primarily rely on the rapid responsiveness of their membranes to address acute inflammation and infections [114]. The neutrophil membrane is enriched with proteins such as integrins and selectins, which are crucial for cell migration and cytokine release, enabling neutrophils to rapidly respond to infections and migrate to inflammatory sites [96]. The neutrophil membrane can recognize tumor cells not only via VCAM-1 expressed on the surface of tumor cells but also through the Fas-Fas ligand interaction. [115,116,117]. Beyond binding to tumor cells [118,119], surface receptors on the neutrophil membrane can recognize tumor cells via the antibody-dependent cell-mediated cytotoxicity (ADCC) mechanism [120].

Both in vitro and in vivo experimental results have shown that the encapsulation of the neutrophil membrane significantly enhances the tumor-targeting ability and accumulation in tumor cells of the nanoplatform, achieving precise strikes on tumor cells. Fan et al. constructed nanoneutrophils (NMPC-NPs) by loading hQ-PTX2 and photosensitizer (Ce6) onto PLGA particles (NPs) with a neutrophil membrane coating [121]. The results showed that the coating of the neutrophil membrane enabled NMPC-NPs to target and accumulate more effectively in tumor tissues, thereby enhancing the drug delivery efficiency and therapeutic effect. Zhang et al. developed another nanoplatform PAM, which combines a neutrophil membrane, silver nanoparticles (AgNPs), and a porphyrin porous coordination network (PCN) [122]. The upregulation of intercellular adhesion molecule-1 (ICAM-1) in tumor cells can mediate neutrophil targeting, so the neutrophil membrane PAM also has better targeting to tumor tissues. Moreover, Qin et al. and Xu et al. constructed nanoplatforms using neutrophil membranes [123], which not only maintained the functions and inflammatory responsiveness of neutrophils but also selectively released drugs at tumor inflammatory sites [124].

Collectively, these studies highlight the significant potential of neutrophil membranes in nanomedicine delivery. Encapsulation with neutrophil membranes not only confers targeted delivery capabilities to inflammatory environments and tumors but also induces cell necrosis and inflammatory responses through phototherapy, thereby enhancing drug recruitment to tumor sites. Furthermore, neutrophil membrane encapsulation effectively prevents drug clearance by the reticuloendothelial system, increases drug blood concentration and circulation time, promotes drug accumulation in tumor tissues, and reduces adverse reactions.

### 4.3. Other Immune Cell Membranes

The coating of T-cell membranes can facilitate the penetration of the blood–brain barrier (BBB) by regulating the expression of tight junction protein ZO-1, temporarily disrupting the integrity of the BBB and constructing efficient penetration pathways for nanoparticles, achieving efficient and safe BBB crossing and demonstrating great potential in the treatment of brain tumors. T-cell membrane recognition of tumor cells is achieved through the binding of TCR to MHC molecules on the tumor cell surface, and this mechanism is universally applicable to all types of tumors [125]. The CD133 and EGFR-specific antibody fragments on the T-cell membrane can enable the embedded nanomaterials to precisely recognize glioblastoma (GBM) tissues. Additionally, through gene editing, researchers can regulate the proteins on the T-cell membrane to adjust its biological properties, thus providing more possibilities for nanomaterials. Wang et al. fabricated CM@AIE NPs [126], taking advantage of this unique property of T-cell membranes, significantly enhancing the targeting ability of nanoparticles to GBM cells and stem cells. Ma et al. constructed GPC 3-CAR lentivirus technology to generate CAR-T cells targeting GPC 3 and extracted the cell membranes (CM) from them to coat nanoparticles, forming tumor-specific nanoparticles (CIM) [127]. Compared with nanoparticles without T-cell membrane coating, CIM demonstrated significantly enhanced tumor targeting ability.

The coating of dendritic cell (DC) membranes also provides excellent tumor targeting ability for nanomedicine delivery platforms. Shi et al. prepared DC cell membrane-coated zinc phosphate nanoparticles (LDC@ZnP NPs) [128], loading colon cancer antigen peptide Adpgk and photosensitizer melanin into the nanoparticles and achieving efficient loading through Zn^2+^ ion chelation. The surface of the vaccine was covered with lipids and DC cell membrane proteins, enhancing its targeting ability to DC cells and promoting lymph node drainage.

Although myeloid-derived suppressor cells (MDSCs) are pathologically activated neutrophils and monocytes with strong immunosuppressive activity [129] and play a prominent role in tumor angiogenesis, drug resistance, and promoting tumor metastasis [130], their active targeting ability to the tumor microenvironment and immune evasion capabilities make them promising for application in nanomedicine. Lan et al. isolated the MDSC membrane from the femur of mice and embedded it on the periphery of black phosphorus (BP) loaded with decitabine through ultrasonic treatment to prepare BP@Decitabine@MDSC, which was named BDM [131]. The MDSC membrane enhanced the active tumor targeting ability of BDM, significantly increasing its accumulation in tumor tissues compared to BP@Decitabine (BD) without membrane coating. In addition, another study constructed a composite nanoparticle GNR@SiO_2_@MnO_2_@MDSCs (referred to as GSMM for short), which was made by coating gold nanorods with MnO_2_ and wrapping them with MDSCs membranes [73]. Due to the wrapping of MDSC membranes, the immune evasion and tumor targeting capabilities were enhanced. The MDSC membranes prolonged the retention time of GSMM in the blood, providing a guarantee for better enrichment of GSMM at the tumor site. The characteristics of the above-mentioned nano-drug delivery platform wrapped by immune cell membranes are detailed in Table 1.

The coating of immune cell membranes significantly improves nanomedicines. By leveraging this approach, nanomedicines can absorb some characteristics of immune cells, thereby unlocking two key skills: first, they are like having an “invisible shield”, skillfully evading recognition and clearance by the body’s immune system and traveling freely in the body; second, they are like being equipped with a precise “tumor navigator”, possessing the outstanding ability to actively search for and target tumor cells. With these two major advantages, nano-drugs can be highly concentrated in tumor tissues while ensuring that normal tissues are not disturbed, significantly reducing the occurrence rate of adverse reactions. Compared with red blood cell membrane and other biological membrane encapsulation strategies, immune cell membrane encapsulation demonstrates unique superiority, opening up new paths for the optimization of nano-drugs and tumor treatment.

## 5. The Immune Cell Membrane Endows the Nano-Photosensitizer with Additional Immune Properties

The immune cell membrane, leveraging its surface proteins, can confer nano-photosensitizers with specific immune functionalities. Moreover, due to the absence of downstream signal transduction pathways, it can effectively serve as an immune modulator without eliciting further cellular responses. Additionally, as a biological membrane, it can be engineered using techniques such as gene editing to meet specific requirements [132,133]. Therefore, encapsulating nano-photosensitizers within immune cell membranes endows the materials with immune characteristics that can be precisely tailored to the tumor microenvironment.

Wang et al. utilized M1-like macrophage-derived extracellular vesicles (M1 EVs) as carriers to encapsulate nanomedicines [134]. By leveraging the properties of the macrophage membrane, they induced M2 macrophages in the tumor microenvironment (TME) to polarize towards the M1 phenotype, significantly enhancing the efficacy of immunotherapy. Yong et al. encapsulated nanomedicines within macrophage membranes [135], taking advantage of the natural signaling molecules (such as TLR4) carried on the macrophage membranes to interact with the tumor microenvironment, thereby promoting the polarization of M2-type tumor-associated macrophages to the anti-tumor M1 type, further activating the immune response, releasing pro-inflammatory cytokines, and inhibiting tumor growth. Du et al.’s research further demonstrated the potential of macrophage membranes in immune regulation [69]. By coating nanomedicines with macrophage membranes, they achieved precise modulation of the immune system. Specifically, the macrophage membrane’s ability to bind and clear endotoxins inhibited the secretion of pro-inflammatory cytokines while promoting the expression of anti-inflammatory cytokines and the polarization of M2 macrophages. Macrophage membrane-coated nanoparticles induce immunogenic cell death (ICD) and activate the cGAS-STING pathway, not only effectively inhibiting tumor growth but also stimulating a strong anti-tumor immune response, including promoting dendritic cell maturation and T-cell proliferation, as well as forming immune memory.

Neutrophil membranes were used to embed iron oxide (Fe_3_O_4_) cores and titanium dioxide (TiO_2_) shells to form nanoparticles (Fe_3_O_4_@TiO_2_ NPs), constructing Neu-FTO [70]. Neu-FTO not only maintained neutrophil activity but also enhanced the expression of immune markers (CD14 and TLR4) and the secretion of cytokines (TNF-α, IL-6, and IFN-γ). This resulted in stronger phagocytic and antibacterial activities, greater chemotaxis to infection sites, and significantly improved mouse survival rates by controlling infection spread and preventing systemic dissemination.

Dendritic cells (DCs) possess the unique ability to recognize tumor antigens expressed on the surface of tumor cells and process these antigens into antigen peptide-major histocompatibility complex (pMHC) forms for precise presentation on their cell surface. Mature DCs can activate and guide different antigen-specific T-cell subsets to precisely target and eliminate homologous tumor cells. The construction of super artificial dendritic cells (saDCs) has opened new avenues for immune regulation. Sun et al. engineered 4T1 cell membranes [136] to express CD86 and anti-LAG-3 proteins, creating saDCs, and coated these membranes on FS-loaded nanoparticles. This innovative approach successfully blocked the immunosuppressive MHC-II/LAG3 pathway, preventing T-cell exhaustion, restoring T-cell activity, promoting rapid proliferation and full activation, and significantly increasing the killing efficiency of T cells against tumor cells. Additionally, it enabled T cells to cross vascular barriers and penetrate the tumor microenvironment, enhancing tumor-targeted delivery efficiency. Importantly, this method can convert “cold tumors” into “hot tumors”, triggering a strong anti-tumor response. In vivo experiments showed that it could effectively inhibit tumor growth, increase survival rates, and induce long-term anti-tumor immune memory effects. Hybrid cell membranes formed by fusing dendritic cells (DCs) with cancer cells retain most of the membrane protein characteristics of both DCs and cancer cells, especially the function of DC cell membranes in recognizing and presenting tumor antigens [137]. DC cell membrane-coated nanomedicines offer multiple advantages for tumor treatment, including enhanced tumor targeting, promotion of immune activation and regulation, increased drug bioavailability, reduced side effects, and synergistic therapeutic effects. These breakthrough achievements highlight the promising application prospects of immune cell membrane-coated nanomedicine delivery systems.

The encapsulation of nanomedicines with T-cell membranes and NK cell membranes also confers unique immune regulatory effects. The binding of PD-1 on T-cell membranes to PD-L1 on tumor cell surfaces significantly promotes the uptake and accumulation of PHD@PM by tumor cells [138]. Additionally, T-cell membranes enhance the immune response by stimulating dendritic cell maturation and activating effector T cells, thereby significantly enhancing anti-tumor immunity. This immune enhancement is partly attributed to the promotion of DC cell maturation by PD-1 expression and the immunogenic cell death effect induced by PDT. NK cells play a unique role in anti-tumor immunity. Unlike T cells, they can spontaneously clear target cells without antigen-specific stimulation or major histocompatibility complex restriction [139]. They regulate immune responses by secreting cytokines such as TNF-α [140,141] and promote the maturation of antigen-presenting cells (APCs) to activate T cells for tumor cell killing [142]. The NK cell membrane is enriched with a diverse array of proteins that can directly recognize tumor cells [125,143]. Deng et al. prepared NK-NP by embedding nanoparticles with NK cell membranes [76]. The membranes confer tumor-targeting ability, enabling precise recognition and DNAM-1 and NKG 2D on NK cell accumulation in tumor tissues, while improving nanoparticle circulation time in the blood. Moreover, NK cell membranes can induce the polarization of pro-inflammatory M1 macrophages, driving the phagocytosis of dying cancer cells and inducing immunogenic cell death, thereby enhancing the antigen presentation process and strengthening NK cell membrane immunotherapy efficacy. The characteristics of the above-mentioned nano-drug delivery platform wrapped by immune cell membranes are detailed in Table 2.

## 6. Nano-Photosensitizers Embedded in Immune Cell Membranes for Synergistic Therapy Against Tumors

In the treatment of unresectable cancers, single chemotherapy, PDT, or immunotherapy often fails to achieve satisfactory therapeutic effects due to certain limitations of the treatment methods or drugs. Therefore, it is crucial to conduct synergistic therapy to complement each other’s deficiencies and achieve a stronger killing effect on tumor cells. In the design and research of nanomedicines related to photosensitizers encapsulated in immune cell membranes, most attempts have been made for synergistic therapy.

PDT therapy mainly works by activating photosensitizers with laser light and is widely used in skin diseases and tumor diseases. The advantage of photodynamic therapy lies in its ability to precisely exert photodynamic effects by controlling laser irradiation. However, the current issue is the low efficiency of photosensitizer enrichment at the target site, which limits the efficacy of photodynamic therapy. Encapsulating photosensitizers in immune cell membranes not only enhances the immunomodulatory effect of PDT but also enables the simultaneous encapsulation of other drugs to achieve the synergistic effect of photodynamic therapy and other drugs (Figure 5). Furthermore, research has consistently shown that compared with membrane-free nanomedicines, nanomedicines encapsulated with immune cell membranes exhibit equivalent or significantly better biocompatibility, with minimal adverse effects on normal tissues and cells. Additionally, the synergistic effect of drug encapsulation and the enhanced targeting ability conferred by immune cell membranes has led to significantly enhanced cytotoxicity, cell activity inhibition, and ROS generation against tumor cells for such nanomedicines.

Wang et al. integrated IR 780 and the NO donor diazeniumdiolate (NONOate) by encapsulating them within macrophage membranes [109], addressing the issue of low accumulation of photosensitizers and active nitrogen drugs in tumor cells. By leveraging the heat-responsive release feature of NONOate, NO gas was released upon laser irradiation and heating, synergistically reacting with ROS (superoxide anions and hydroxyl radicals) generated by the photosensitizer. This reaction formed the highly toxic peroxynitrite anion ONOO−, thereby enhancing the synergistic effect of photodynamic therapy (PDT) and nitric oxide-based therapy, significantly increasing toxicity to breast cancer cells. In another study, Wang et al. embedded Cu and photosensitizers within macrophage membranes [103], further increasing ROS levels in the tumor microenvironment by depleting GSH with Cu^2+^, thereby enhancing the cytotoxicity of PDT against tumors. Zhao et al.’s nanomedicine delivery platform [73] utilized Mn^2+^ to catalyze the conversion of H_2_O_2_ into ROS for chemodynamic therapy (CDT), activating the cGAS-STING pathway. Moreover, Mn^2+^ directly stimulated STING to induce the secretion of IFN-I, pro-inflammatory cytokines, and chemokines, enhancing anti-tumor immunity. Additionally, immunogenic cell death (ICD) induced by CDT and photothermal therapy (PTT) further boosted anti-tumor immunity, achieving a synergistic promoting effect.

Macrophage membranes with M1 inflammatory phenotypes can induce ICD, activate antigen-presenting cells, and stimulate the production of tumor-specific effector T cells in metastatic tumors, significantly enhancing anti-tumor immune efficiency [144]. Therefore, the persistent immune response elicited by the photosensitizer nanoplatform embedded in macrophage membranes can inhibit the rebound of primary tumors after PDT and enhance the efficacy of immunotherapy on primary tumors, leading to regression of distant non-irradiated tumors, achieving a synergistic effect of PDT and immunotherapy. Fang et al. achieved combined immunotherapy and PDT by embedding siRNA/ICG in macrophages [145]. The siRNA targeting PD-L1 (siPD-L1) weakened the expression of immunosuppressive PD-L1 induced by PDT, effectively triggering a robust anti-tumor immune response in a “self-synergistic” manner. Photosensitizers induce ICD under light exposure, releasing damage-associated molecular patterns (DAMPs), enhancing APC presentation, promoting CTL infiltration, and improving the tumor microenvironment. Tumor-associated antigens generated by PDT can elicit systemic anti-tumor immune responses but require immune system activation techniques to assist. The targeting and immune characteristics of immune cell membranes can compensate for this deficiency. Meng et al. constructed M@PFC [105], embedding the photosensitizer PF3-PPh3 and the immunomodulator CpG in macrophage membranes. After targeting tumor cells with macrophage membranes, it stimulated cellular immunity in tumors. Besides using photodynamic therapy to kill tumor cells, it also promoted cytotoxic T-cell responses through the immunomodulator CpG, significantly enhancing the cytotoxic effect of photodynamic therapy on tumors.

In Zhang et al.’s study [138], combining ICD induced by PDT with PD-L1 antibody treatment for immune checkpoint blockade (ICB) significantly enhanced the anti-tumor immune response, achieving a synergistic effect of photodynamic and immunotherapy. This therapy effectively inhibited the progression of primary and metastatic tumors by promoting dendritic cell maturation and CTL tumor infiltration. Sun et al. used super artificial dendritic cell membranes to embed Fs-NP nanomedicines [136]. Besides exerting photodynamic effects, it utilized the co-expression of CD86 and anti-LAG-3 on super artificial dendritic cell membranes to activate T cells, achieving a combined effect of immunotherapy and PDT. The dual-signal mechanism of dendritic cells promotes T-cell activation and proliferation, and simultaneously restores T-cell exhaustion through LAG 3 antibodies, thereby stimulating a strong tumor-specific T-cell response. The characteristics of the above-mentioned nano-drug delivery platform wrapped by immune cell membranes are detailed in Table 2.

## 7. Immunocyte Membrane Combined with Photodynamic Therapy for Precise Regulation of Chemotherapy

Chemotherapy is a crucial step in tumor treatment. Postoperative chemotherapy for tumor patients can effectively prevent tumor recurrence. However, due to the lack of selectivity of chemotherapy drugs, they not only kill tumor cells but also a large number of normal cells, leading to various side effects in patients. Under the wrapping effect of immunocyte membranes, the tumor targeting problem of chemotherapy drugs is largely solved, reducing the damage to normal body tissues. Moreover, the presence of immunocyte membranes enables the drugs to have immunomodulatory capabilities, enhancing the killing effect on tumor tissues. When immunocyte membranes combine chemotherapy drugs with phototherapy, they can control the timing and conditions for chemotherapy to take effect, achieving precise regulation of chemotherapy while exerting a synergistic therapeutic effect. Furthermore, under the wrapping effect of immune cell membranes, the drug targeting issue has been significantly improved, enabling the effective synergistic treatment of PTT and chemotherapy [108,123,131].

M1 macrophage membranes were used to embed CPPO and Dox-EMCH [144]. After the macrophage membranes targeted the drugs to the tumor, the reaction between H_2_O_2_ produced by tumor cells and CPPO generated chemical energy that induced the rupture of the M1 macrophage membranes, thereby releasing Dox-EMCH. Subsequently, in the acidic microenvironment of the tumor, Dox-EMCH rapidly activated into toxic doxorubicin (Dox), killing the tumor. Moreover, embedding photosensitizers and AQ4N in macrophage membranes can break through the blood–brain barrier, making it easier for chemotherapy drugs to reach the brain. The consumption of oxygen by photodynamic therapy and the subsequent conversion of AQ4N into toxic AQ4 in the hypoxic environment of the tumor activated the chemotherapy effect of the drugs [134]. Fan et al. combined hypoxia-responsive PTX dimer prodrug (hQ-PTX) and photosensitizer Ce6 and embedded them. Through PDT to generate ROS and consume oxygen in the tumor tissue, hQ-PTX was degraded to achieve specific release of PTX in the tumor tissue, not only enhancing the killing effect on the tumor but also reducing the side effects of paclitaxel [121]. Cisplatin is a common chemotherapy drug, and platinum itself is a good catalyst. Liu et al. used macrophage membranes to embed platinum nanoparticles and photosensitizers. The catalytic supply of platinum nanoparticles enhanced the PDT effect, and PDT-mediated membrane permeabilization enabled nano-Pt to better penetrate tumor cells to enhance the chemotherapy effect, achieving a bidirectional synergistic effect of chemotherapy and PDT [107]. Hu et al. used DC cell membranes to incorporate doxorubicin (DOX) and indoleamine 2,3-dioxygenase 1 inhibitor (IDO1). This reversed the inhibitory tumor environment and facilitated the generation of anti-tumor cytotoxic T cells. By increasing the uptake of tumor-derived antigens and their presentation to T cells through DCs, the anti-tumor immunity was enhanced, achieving a synergistic effect of chemotherapy and immunotherapy [102].The characteristics of the additional nano-drug delivery platforms encapsulated by immune cell membranes are detailed in Table 3.

## 8. Perspectives

The current landscape of cancer treatment primarily revolves around surgery and chemotherapy. However, in many cases, when cancer patients are found, the disease is already at an advanced stage, and surgery and chemotherapy alone are no longer effective in prolonging the survival of patients. There is an urgent need to develop new therapies (e.g., two or more combination therapies) and add adjuvant therapies (e.g., phototherapy, immunotherapy, etc.) to completely eradicate tumors. In combination therapy (e.g., PDT/PTT/chemotherapy, PDT/chemotherapy/immunotherapy, PDT/chemotherapy, etc.), Phototherapy, in particular, has emerged as a safe, easily controllable, and non-invasive adjuvant therapy for cancer treatment. The efficacy of phototherapy heavily relies on the accumulation of photosensitizers in tumor tissues and their anti-tumor effects, underscoring the importance of developing potent photosensitizers to enhance the efficacy of phototherapy.

While advancements have been made in improving photosensitizer design to achieve high drug loading rates and enhanced tumor targeting, challenges such as poor permeability, tumor hypoxia, and low specificity still persist. These challenges often necessitate the use of high doses of photosensitizers, which may lead to prolonged side effects and poor photosensitivity in patients. To address these issues, researchers have developed triplet photosensitizers with ultra-high efficiency to reduce the required dosage of photosensitizers without compromising therapeutic efficacy [149]. The high temperature of PTT may increase the damage to normal tissues as well as the heat resistance of tumor tissues, so a rational design of PTT and photosensitizer that can be achieved at low temperatures has been developed [150,151]. Exploration of the near-infrared (NIR-II) window has opened new avenues in photosensitizer research, offering a promising direction for the development of future photosensitizers with enhanced properties. Furthermore, improving drug delivery systems (DDS) to enhance tumor targeting, optimize drug release in tumor tissues, and minimize off-target effects on normal tissues remains a critical focus area for improving treatment outcomes. These studies will continue to advance the treatment of tumors and provide patients with better treatments to prevent the progression of the disease.

Therefore, immune cells and immune cell membranes have attracted great attention due to their unique biological characteristics and the abundant functional groups on the surface of their membranes (Table 4). Macrophage membranes, for instance, facilitate the targeted accumulation of therapeutic agents in tumor tissues due to the presence of specific chemokines or chemokine receptors on their surfaces. M1EVs have been shown to promote the transition from M2 to M1 macrophages, which are capable of engulfing tumor cells and generating ROS, such as H_2_O_2_. This ROS production can enhance the efficacy of PDT, induce ICD, and stimulate anti-tumor immune responses. The synergistic effect of PDT and tumor immunotherapy is markedly enhanced by the use of these macrophage-derived materials. Neutrophil membrane nanomaterials, containing most of the surface proteins of neutrophils, exhibit the potential to target inflammatory tumor environments and accumulate drugs in tumor tissues, similar to neutrophils. The wrapping of these materials with neutrophil membranes enables evasion of immune system clearance, ensuring prolonged circulation in the body and excellent biocompatibility. By encapsulating various drugs, these nanomaterials create favorable conditions for combining multiple treatment modalities, significantly enhancing the synergistic anti-tumor effects of PDT in conjunction with immunotherapy and chemotherapy. Dendritic cell (DC) membrane nanomaterials contribute to nanoparticle stability while retaining surface proteins that can effectively interact with T cells. This interaction mediates T-cell proliferation and activation, thereby supporting tumor immunotherapy. Targeting tumor cells via T cells allows the nanoparticles carried by T cells to penetrate biological barriers, traverse the tumor microenvironment, and efficiently accumulate in tumor tissues, achieving high drug targeting specificity and enhanced delivery efficiency. The coating of T-cell membranes significantly augments the capacity of nanomedicines to penetrate the blood–brain barrier, thereby facilitating more efficient drug delivery. This feature provides T-cell membranes with a distinct advantage over other biological membranes in the context of brain tumor treatment. By leveraging lentiviral DNA transfection technology, T-cell membranes can be engineered to specifically target tumor-associated surface molecules, further enhancing their recognition and binding efficiency to tumor tissues. This strategy not only significantly improves the targeting of nanomedicines but also provides new ideas for the future development of biological membrane-coated nanomedicine delivery systems. Moreover, T-cell membranes enhance immune responses through the stimulation of dendritic cell maturation and activation of effector T cells, thereby significantly amplifying the anti-tumor immune response. Natural killer (NK) cell membrane nanomaterials not only improve nanoparticle stability and biocompatibility but also hold promise in tumor targeting through the recognition of tumor-specific ligands by the proteins on NK cell membranes. Additionally, NK cell membranes can induce pro-inflammatory M1 macrophage polarization, enhancing the immunotherapy effect on tumor cells through a cascade of subsequent reactions. MDSCs are recruited to tumor tissues by chemokines secreted by tumor cells, thereby exhibiting a targeting effect on tumors. Coating nanomedicines with MDSC membranes not only confers specific tumor-targeting capabilities but also retains the inherent ability of MDSCs to evade host immune clearance. This dual functionality significantly prolongs the circulation time of nanomedicines in the bloodstream and enhances their accumulation at tumor sites. We believe that, based on current research, macrophage membranes are considered the most suitable carriers for photosensitizers among immune cell membranes due to their natural tumor chemotactic ability, ROS generation induced by M1 polarization, and synergistic immune regulatory effects. Macrophage membranes have demonstrated highly efficient synergy in PDT/chemotherapy/immunotherapy in models such as glioma and breast cancer, achieving a relatively high level of technological maturity. T-cell membranes have irreplaceable advantages in the treatment of brain tumors, while NK cell membranes show significant potential in controlling metastatic foci, and can be used as supplementary options in specific scenarios. In the future, their adaptability needs to be further enhanced through engineering modifications (such as targeted modification of membrane proteins and hybrid membrane design).

The use of immune cell membranes to encapsulate and load therapeutic drugs such as photosensitizers is expected to significantly extend their circulation time in vivo and have good biocompatibility, enhance their accumulation in tumor tissues, and indicate the possible enhancement of PDT and PTT and improve the therapeutic effect of immunotherapy. Therefore, synergistic therapy using immune cells and immune cell membrane nanoparticles as carriers for combined phototherapy, chemotherapy, and immunotherapy shows promise. As shown in the paper, the drug delivery system encapsulated by the immune cell membrane achieves high tumor targeting efficiency. It significantly increases circulation time in the body, as well as the accumulation and release of photosensitizers and other drugs in tumor tissues, and also has excellent effects on synergistic anti-tumor immunotherapy. However, in the aforementioned studies, the influence of the changes in average particle size, zeta potential, and polydispersity index (PDI) caused by the immune cell membrane coating of nanomedicines has not been fully explored. Instead, these physical and chemical property changes were merely regarded as indicators of successful preparation of immune cell membrane-coated nanomedicines. This aspect needs to be given due attention in future research.

.

The development of novel DDSs based on immune cell membranes presents exciting opportunities for advancing cancer treatment through synergistic approaches such as phototherapy and immunotherapy. However, several challenges, such as the potential rejection of allogeneic immune cells, high fabrication costs of DDSs, lack of standardized isolation and purification techniques, and complexities in large-scale clinical manufacturing, hinder the widespread implementation of these cutting-edge technologies. Therefore, technologies that reduce the cost of immune cell membrane nanoparticles and increase their large-scale production, so that they can stably load different drugs and photosensitizers and can achieve synergies in PDT or PTT and other adjuvant therapeutics, are promising areas [45].

In addition, no relevant research progress has been found on the use of B lymphocyte membrane-encapsulated nanoparticle delivery systems. Although there have been some explorations of NK cell membrane and T-cell membrane-encapsulated nanoparticle delivery systems in recent years, the related research is still relatively limited. These fields urgently need further expansion and improvement, and they show broad application prospects in the future in the research and development of immune cell membranes. In addition, multi-membrane hybrid strategies (such as combining the macrophage-targeting function with the T-cell immune activation advantage) are expected to become a highly promising development direction in the future, which is worth in-depth exploration. We firmly believe that the development of high-performance photosensitizers and extensive research on immune cells may be a promising approach in phototherapy and anti-tumor immunotherapy, and achieving the synergistic effect of PDT/chemotherapy/immunotherapy will be a promising way to treat cancer diseases in the future.

## Figures and Tables

**Figure 1 pharmaceutics-17-00481-f001:**
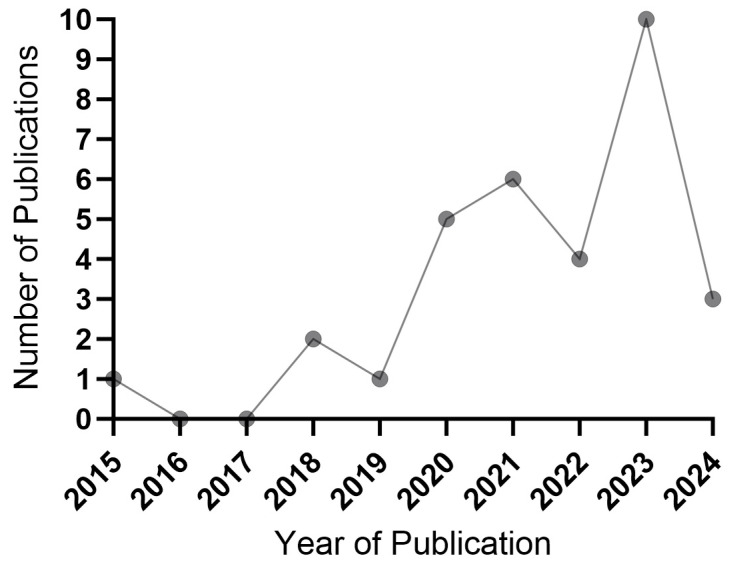
In recent years, the number of studies on nano-photosensitizers embedded in immune cell membranes has shown an upward trend.

**Figure 2 pharmaceutics-17-00481-f002:**
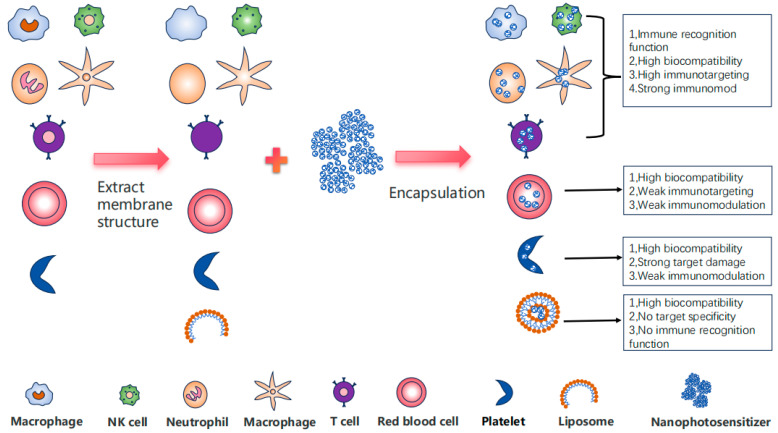
The comprehensive advantages of immune cell membrane-embedded nanomaterials. The encapsulation of nanomaterials by biological membranes mainly consists of two steps: the preparation of the biological membrane and the wrapping of the material. The presence of biological membranes typically enhances the biocompatibility of nanomaterials and imparts unique biological characteristics to the materials. Specifically, the encapsulation of nanomaterials with immune cell membranes can confer certain immunological properties to the materials.

**Figure 3 pharmaceutics-17-00481-f003:**
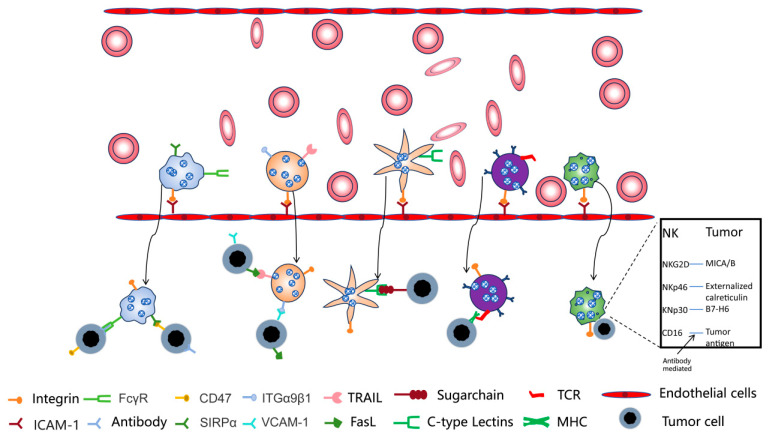
The basis of immune cell recognition of tumors. Adhesion molecules are present on immune cell membranes, which can bind to ligands on vascular endothelial cells in the human body, allowing the cells to migrate to tumors. Subsequently, tumor cells can continuously interact with various proteins on different immune cell membranes through molecular interactions. Based on these factors, after encapsulating nanomaterials with immune cell membranes, the biological properties of the immune cell membranes can be utilized to target tumors.

**Figure 4 pharmaceutics-17-00481-f004:**
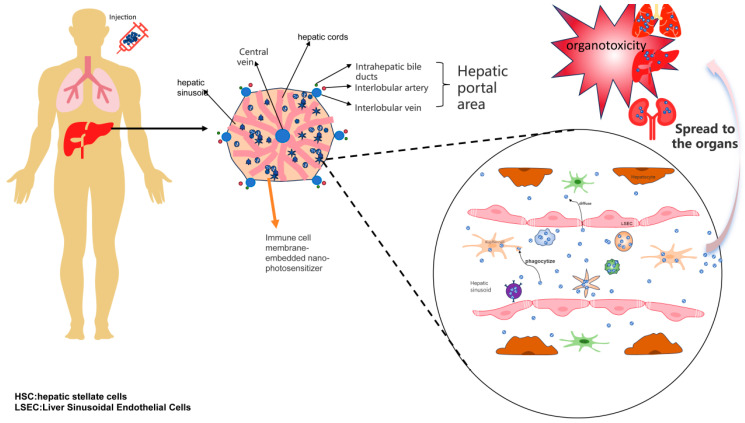
Immune cell membrane-wrapped nano-photosensitizer enhances material stability and reduces organ toxicity. Due to their small size, nanomaterials can easily penetrate various biological barriers and enter different organs in the body, which may lead to damage to normal organs. Additionally, as nanomaterials are recognized as foreign substances, various types of macrophages in the liver can readily phagocytize and eliminate them. However, after encapsulating the nanomaterials with immune cell membranes, the presence of immune cells allows the nanomaterials to evade these issues by utilizing the biological properties of the immune membranes.

**Figure 5 pharmaceutics-17-00481-f005:**
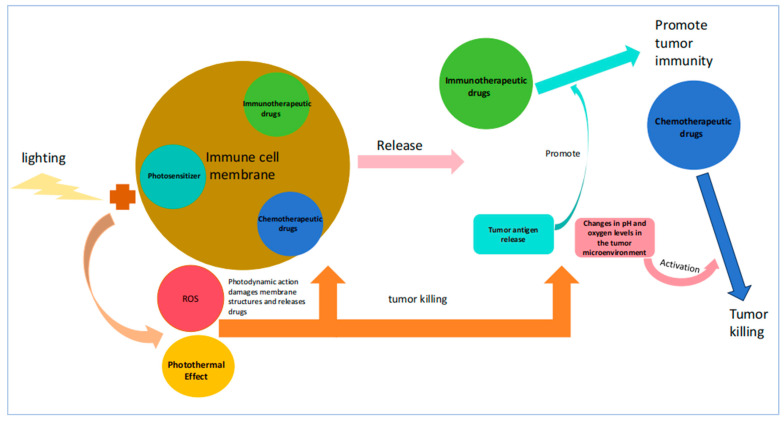
Schematic illustration of the synergistic mechanism of nano-photosensitizers encapsulated within immune cell membranes in combination with other therapeutic agents.

**Table 1 pharmaceutics-17-00481-t001:** Characteristics of delivery systems for enhancing targeting efficiency and drug stability.

Investigators	Nanoplatform	Average Particle Size (nm) (±S.D.)	Zeta Potential (mV) (±S.D.)	PDI	Photosensitizer	Membrane Structure	Other Drugs	Synergistic Treatment Modalities	Target Disease
Gao et al. [99]	CMA nPS	~200	~15	0.262	DBCO-TAPP	azide-modified macrophage cell membrane with a VSV-G-modified NIH3T3 cell membrane	--	PDT + Ca^2+^ overload	Pulmonary carcinoma
Xie et al. [100]	HB-NLG8189@MPCM	232.46 ± (6.52)	−26.96 ± (4.02)	--	chlorine6-C15-ethyl ester (HB)	Macrophage membrane	indoleamine-(2,3)-dioxygenase (IDO) pathway inhibitor	SDT + immunotherapy	Triple-Negative Breast Cancer
Sun et al. [101]	AuND-TPP-ICG@MCM	135.1 ± (3.0)	−3.3 ± (0.1)	--	indocyanine green	macrophage cell membrane	multifunctional gold nanodendrite; triphenylphosphonium	PDT + PTT	Breast cancer
Hu et al. [102]	(C/I) BP@B-A (D) &M1m	117.0 ± (7.5)	--	--	Chlorin e6	M1 macrophage cell membranes	Doxorubicin; indoleamine 2,3-dioxygenase 1 inhibitor	PDT + Chemotherapy + ICD	Breast cancer; Cutaneous melanoma
Wang et al. [103]	M-Cu-T	172 ± (1)	−11 ± (1)	--	meso-tetra (4-aminophenyl) porphyrin	M1-Raw264.7 cell membranes	Cu^2+^	PDT + immunotherapy	Colon cancer; Lung cancer
Chen et al. [104]	NPR@TAMM	91 ± (4)	−20	--	NaYF4:Yb	tumor-associated macrophage membrane	Rose Bengal (NPR)	PDT + ICD	Breast cancer
Meng et al. [105]	M@PFC	200	~−25	--	PF3-PPh3	Macrophage membrane	immune adjuvant (CpG)	Immunotherapy + PDT	Breast cancer
Zhao et al. [106]	M@BS-QE NP	155.3	−19.1	0.312	bismuth selenide nano-Particles	Macrophage membrane	quercetin	PTT + Chemotherapy	Breast cancer
Liu et al. [107]	nano-Pt/VP@MLipo	140	−16.7	--	verteporfin	RAW264.7 macrophage brane	Platinum Nanoparticles	Chemophototherapy	Breast cancer
Yu et al. [108]	CuSCDB @ MMT 7	222.5 ± 20.1	−18± 1.8	--	CuSCDs	macrophage membrane hybridized with T7 peptide	Bortezomib	PTT + Chemotherapy	Breast cancer
Wang et al. [109]	IR 780-NO-PFH-Lip@M	--	--	--	near-infrared fluorescent dye(IR780)	macrophage cell membranes	Diazeniumdiolate; perfluorocarbon	reactive nitrogen species therapy + PTT/PDT	Breast cancer
Fan et al. [121]	NMPC-NPs	165	−12.6	--	Chlorin e6	neutrophil membrane	Paclitaxel (PTX) dimeric prodrug	PDT + Chemotherapy	Breast cancer
Zhang et al. [122]	PAM	220	−15	--	porphyrinic porous coordination network	neutrophil membrane	silver nanoparticles	PDT + metal ions therapy	Colon cancer
Qin et al. [123]	I-L@NM	61 ± (6)	−10	--	indocyanine green	neutrophil membrane	β-Lapachone;	PTT + Chemotherapy	Colon cancer
Xu et al. [124]	PAN	91.25 ± (0.34)	−40.21 ± (3.12)	--	Chlorin e6	neutrophil membrane	cationic RGD-apoptotic peptide conjugate	PDT + Chemotherapy	Squamous cell carcinoma of skin; Tongue squamous cell carcinoma
Wang et al. [126]	CM@AIE NPs	107	7	--	AIE-gens	genetically engineered CAR T-cell membrane	--	PTT	glioblastoma
Ma et al. [127]	CIM	110	−6.7	0.288	near-infrared fluorescent dye (IR780)	GPC3 targeting CAR-T-cellmembranes	mesoporous silica nanoparticles	PTT	Hepatocellular carcinoma
Shi et al. [128]	LDC@ZnP NPs	30	−10	--	Melanin	dendritic cell membrane	Adpgk, zinc phosphate nanoparticles	PTT + immunotherapy	Colon cancer
Lan et al. [131]	BDM	264	−23.5	--	Black phosphorous	myeloid-derived suppressor cell membrane	Decitabine	PTT + PDT + Chemotherapy	Oral squamous cell carcinoma
Zhao et al. [73]	GNR@SiO_2_@MnO_2_@MDSCs (GSMM)	129	−35.43	--	Gold nanorod	myeloid-derived suppressor cells membrane	MnO_2_	PTT+ CDT	Cutaneous melanoma; Breast cancer

PDI: Polymer dispersity index; DBCO-TAPP: dibenzocyclooctyne-conjugated meso-tetra(4-aminophenyl) porphyrin; VSV-G: Vesicular stomatitis virus glycoprotein; PDT: photodynamic therapy; SDT: sonodynamic therapy; PTT: photothermal therapy; ICD: immunogenic cell death; NaYF4:Yb: rare-earth-upconversion-nanoparticle(UCNP)-based PS; PF3-PPh3: aggregation-induced emission photosensitizer; CuSCDs: CuS nanoparticles composited with carbon dots; AIE-gens: aggregation-induced-emission (AIE)-active luminogens; MnO_2_: manganese dioxide; CDT: Chemo-dynamic therapy.

**Table 2 pharmaceutics-17-00481-t002:** Characteristics of delivery systems for enhancing targeting and providing immunomodulation.

Investigators	Nanoplatform	Average Particle Size (nm) (±S.D.)	Zeta Potential (mV) (±S.D.)	PDI	Photosensitizer	Membrane Structure	Other Drugs	Synergistic Treatment Modalities	Target Disease
Wang et al. [134]	CCA-M1EVs	100	--	--	chlorin e6	M1-like macrophage-derived extracellular vesicles	CPPO; banoxantrone	CDT + Chemotherapy	Glioblastoma multiforme
Yoon et al. [135]	UCNPs@mSiO_2_PFC/Ce6@RAW-Man/PTX	61.3 ± 1.1	−11.6	--	chlorin e6	macrophage membranes	perfluorocarbon; paclitaxel	PDT + immunotherapy	Breast cancer
Du et al. [69]	MCeC@MΦ	71.2 ± (1.9)	−40	--	chlorin e6	macrophage membranes	cerium oxide nanocatalyst	PDT + immunotherapy	Multi-drug-resistant bacterial sepsis
Zhang et al. [70]	Neu-FTO	300	15	--	TiO_2_	neutrophil membrane	Fe_3_O_4_	PDT + immunotherapy	Infection
Sun et al. [136]	saDC@ Fs-NP	110 ± (2.5)	−9.35 ± (0.68)	--	AIE photosensitizer (FS)	superartificial dendritic cells membranes	--	PDT + immunotherapy	Breast cancer
Liu et al. [137]	PCN@ FM	~160	−32	0.11	porphyrin-based Zr-MOF (PCN-224)	cytomembranes of hybrid cells acquired from the fusion of cancer and dendritic cells	--	PDT + immunotherapy	Breast cancer
Zhang et al. [138]	PHD@PM	150	−14.5	--	sinoporphyrin sodium	PD-1-expressing HEK293T-cell membranes	human serum albumin-perfluoro-tributylamine nanoemulsion	PDT + immunotherapy	Breast cancer
Deng et al. [76]	NK-NP	80 ± (1.5)	--	0.105	TCPP	natural killer cell membrane.	--	PDT + immunotherapy	Breast cancer

PDI: Polymer dispersity index; CPPO: hydrophobic bis(2,4,5-trichloro-6-carbopentoxyphenyl) oxalate; CDT: chemiexcited photodynamic therapy; PDT: photodynamic therapy; AIE: aggregation-induced-emission; TCPP: 4,4′,4″,4‴-(porphine-5,10,15,20-tetrayl) tetrakis (benzoic acid).

**Table 3 pharmaceutics-17-00481-t003:** Characteristics of other delivery systems loaded with immune cell membranes.

Investigators	Nanoplatform	Average Particle Size (nm) (±S.D.)	Zeta Potential (mV) (±S.D.)	PDI	Photosensitizer	Membrane Structure	Other Drugs	Synergistic Treatment Modalities	Target Disease
Fang et al. [145]	siRNA/ICG@DSeSPm	~116	−13 ± (2)	--	indocyanine green	macrophage membrane	siPDL1	PDT + immunotherapy	Breast cancer
Ding et al. [144]	M1 CCD	165 ± (31)	--	--	chlorin e6	M1 macrophage-derived extracellular vesicles	CPPO	PDT + Chemotherapy	Breast cancer
Cao et al. [146]	EG@EMHMNPs	230 ± (50)	−38.26 ± (0.36)	--	Emodin	EMHM	glycyrrhizin	PDT + Chemotherapy	melanoma
Steen J. Madsen et al. [97]	Ma-AuNS	--	--	--	gold–silica nanoshells (AuNS)	Rat alveolar macrophages membrane (Ma)	--	PTT	Glioma
Zhang et al. [147]	NM-HB NPs	140	−24.8	--	Hypocrellins (HB)	neutrophil membrane	--	PDT	Hepatocellular carcinoma
Xu et al. [148]	DC@AIEdots	113.2	−12.8	0.121	MeTIND-4	dendritic cell membrane	--	PDT + immunotherapy	Breast cancer

PDI: Polymer dispersity index; PDT: photodynamic therapy; CPPO: prodrug aldoxorubicin (Dox-EMCH),bis [2,4,5-trichloro-6-(pentyloxycarbonyl)phenyl] oxalate; EMHM: fused erythrocyte and macrophage to form a hybrid membrane; PTT: photothermal therapy.

**Table 4 pharmaceutics-17-00481-t004:** Comparison of characteristics of immune cell membrane-based drug delivery systems.

Types of Membrane Carriers	Core Functions and Mechanisms	Advantages	Adapt to Diseases	Compatibility	Evaluation
Macrophage Cell Membrane	Targeting chemokine receptors (such as CCR2/CXCR4) in the tumor microenvironment; inducing M2 to M1 macrophage polarization, enhancing ROS production and immune response.	Highly efficient tumor targeting; Synergistic PDT and immunotherapy; Improving the hypoxic microenvironment of tumors	Solid tumors (such as glioma, breast cancer);Combined PDT/chemotherapy/immunotherapy	★★★★★	Macrophage membranes perform best in enhancing the efficacy of photodynamic therapy (PDT) and reshaping the immune microenvironment, especially for ROS-dependent photosensitizers.
Neutrophil Cell Membrane	Inflammation targeting (CXCR1/CXCR2 receptors); Evading immune clearance, prolonging circulation time; Inducing M1 macrophage polarization	High biocompatibility; Penetrate the inflammatory barrier; Support multiple laser treatments	Infection-related tumors or metastases; PDT combined with antibacterial/anti-inflammatory therapy	★★★★☆	It is suitable for scenarios requiring long cycles and combined multi-mode treatments, but the ability to generate ROS depends on the design of the photosensitizer.
T Cell Membrane	PD-1/PD-L1 blockade reverses T-cell exhaustion; penetrates the blood–brain barrier (BBB); activates the DC-T cell axis	Brain tumor-specific delivery; Immune checkpoint blockade enhancement; Activation of systemic anti-tumor immunity	Glioma, metastatic brain tumor; PDT combined with immune checkpoint inhibitors	★★★★☆	It has unique advantages for brain tumors, but the stability of membrane proteins needs to be optimized to maintain the function of PD-1.
Dendritic Cell Membrane	MHC-I/II and co-stimulatory molecules (CD80/CD86) activate T cells; induce tumor antigen-specific immune responses.	Efficient T-cell activation; natural antigen-presenting function; support the transformation of “cold tumors” to “hot tumors”	Low immunogenic tumors (such as pancreatic cancer); PDT combined with personalized vaccine therapy	★★★☆☆	It needs to be combined with tumor antigen loading technology, is suitable for customized photoimmunotherapy, but has a relatively high preparation complexity.
NK Cell Membrane	NKG2D/DNAM-1 mediates tumor recognition; induces M1 macrophage polarization; and synergizes with PDT to enhance the distant effect.	Innate immune activation; inhibition of tumor metastasis; long-lasting immune memory	Highly metastatic tumors (such as melanoma); PDT combined with adoptive cell therapy	★★★★☆	It has performed outstandingly in the control of metastatic foci, but the issue of large-scale preparation of NK membrane proteins needs to be addressed.
MDSCs Cell Membrane	Tumor chemokine receptor targeting; Evading immune surveillance	High tumor accumulation efficiency; low immunogenicity	Immunosuppressive tumor microenvironment; PDT combined with immunosuppression reversal therapy	★★☆☆☆	The potential remains to be verified. It is applicable to highly immune-escape tumors, but it may aggravate immune suppression and thus requires careful design.

Evaluation criteria for photosensitizer Compatibility: ★★★★★: Optimal, significantly enhances photosensitizer targeting, ROS generation, and immune synergy (e.g., macrophage membrane); ★★★★☆: Excellent, has a clear synergistic mechanism but depends on photosensitizer properties (e.g., T/NK cell membrane); ★★★☆☆: Good, requires additional functionalization design (e.g., dendritic cell membrane needs antigen loading); ★★☆☆☆: Requires further research, potential risks or mechanisms are unclear (e.g., MDSCs membrane).

## Data Availability

No new data were created or analyzed in this study. Data sharing is not applicable to this article.

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
