# Peer review of "Tumor Treatment by Nano-Photodynamic Agents Embedded in Immune Cell Membrane-Derived Vesicles"

_pharmaceutics, 2025, doi:10.3390/pharmaceutics17040481_

Round 1
Reviewer 1 Report
Comments and Suggestions for Authors
This is a review on the use of PDT, with a focus on embedding in cell membranes. The authors particularly discuss the utility of immune cells for this. This is quite niche for a review, but I did find it interesting in some aspects. However, it needs work before it can be considered for publication, particularly I found the text needs refinement and clarity in the points the authors are making.
- On line 43 the authors refer to low toxicity, on line 56 the authors refer to phototoxicity. Needs consistency and clarity.
- language line 62 - prescriptions?
- On line 88 PDT/immunotherapy is presented as "promising". On line 89 the text begins by considering "problems", this again needs clarity to flow more logically.
- language line 99 "include such as"
- line 155 - not sure these are "roles", rather than abilities or functions.
- Some tautology in places. Line 161 "immune cell membranes... originate from... immune cells" - this is obvious, text needs to be less wordy.
- line 168 - may be useful to clarify why immunogenicity is a "risk" - when much immunotherapy is aimed at generated an immune response? Line 184 - also refers to inflammatory response, line 194 "immunogenic reactions" - this is repetitive and review could be more concise.
- line 200 - citation to support immune evasion using red blood cells? Line 225 - citation to support enrichment at tumor sites?
- Section 4.1 is extremely long and wordy. Needs breaking up and possibly making more concise. Line 323 for example uses the phrase "nano-drug delivery platform" twice and could be clearer.
- line 378 not sure what you mean by "green channel"?
- line 516 language "themembranes".
- line 550 check that "NONOate" is defined.
- Line 614 tautology again - "can increase solubility... which is beneficial for increasing their solubility".
- I think the general argument for using immunocyte drugs is weak in line 614-615. Surely embedding in any membrane increases solubility, immunocyte membranes are not unique in this respect? Given that this is key in the article title, this needs more comparison and context.
- The figure legends lack information, they should explain what is being shown in the figures. The figures themselves are reasonably clear and effective but legends should explain them.
As outlined in comments, article is wordy, with some tautology and lack of clarity in places. Needs some refinement and re-structuring.
Author Response
Thank you for your valuable comments and suggestions on our manuscript entitled "Tumor Treatment By Nano-Photodynamic Agents Embedded in Immune Cell Membrane" (ID: pharmaceutics-3516044).
We have carefully revised the manuscript according to the reviewers' feedback. Below, we provide a point-by-point response to all comments. All modifications in the revised manuscript are highlighted in Bold font.We hope these revisions address the concerns raised and improve the quality of the manuscript. Please do not hesitate to contact us if further clarification is needed.
Comments 1
“ On line 43 the authors refer to low toxicity, on line 56 the authors refer to phototoxicity. Needs consistency and clarity.”
Response 1
Thank you for your critical comments. The "low toxicity" mentioned in line 43 of the original manuscript refers to systemic low toxicity, while the "phototoxicity" in line 56 refers to local phototoxicity. We have made the detailed changes at (page 3, line 86).
Comments 2
language line 62 -prescriptions?
Response 2
We appreciate the reviewers' acute observation on the accuracy of the terms. The original text has been corrected from "prescription" to "adverse effects" at that point.
Comments 3
On line 88 PDT/immunotherapy is presented as "promising". On line 89 the text begins byconsidering "problems", this again needs clarity to flow more logically
Response 3
We are grateful to the reviewers for their constructive comments on the structure of the article. A logical transition has been added in Part 1 (lines 128-133 on page 5), providing a clearer logical flow.
Comments 4
language line 99 "include such as"
Response 4
We are grateful to the reviewers for their astute correction of the grammatical redundancy. As requested, we have modified "include such as" to "include" in this section of the first chapter.
Comments 5
line 155 -not sure these are "roles", rather than abilities or functions.
Response 5
We are grateful to the reviewers for their constructive comments. In Section 3 of the revised text (line 258, page 10), "roles" has been changed to "functional capabilities" to ensure an accurate description of the characteristics of cellular mechanisms.
Comments 6
Some tautology in places. Line 161 "immune cell membranes... originate from... immune
cells" _this is obvious, text needs to be less wordy.
Response 6
We are grateful to the reviewers for their editorial suggestions. We have streamlined the redundant expressions in Section 3 of the revised text (line 273 on page 10), eliminating the superfluous source explanations while maintaining scientific rigor.
Comments 7
. line 168 - may be useful to clarify why immunogenicity is a "risk" - when much immunotherapyis aimed at generated an immune response? Line 184- also refers to inflammatory responseline 194 "immunogenic reactions" - this is repetitive and review could be more concise.
Response 7
Thank you very much for your valuable comments! We have further clarified in the immunogenicity section (page 11, line 279) why there is a "risk" here. Meanwhile, the inflammatory response mentioned in line 184 (revised position: page 11, line 299) refers to the possibility that the antigenic components on the bacterial membrane may overly activate the inflammatory response. Additionally, we have refined the expression in line 194 to make it clearer.
Comments 8
line 200 - citation to support immune evasion using red blood cells? Line 225 - citation to support enrichment at tumor sites?
Response 8
Thank you very much for your valuable suggestions! We have respectively supplemented the supporting literature "Cell Membrane-Camouflaged Nanocarriers for Cancer Diagnostic and Therapeutic." and "Immune Cell Membrane-Coated Biomimetic Nanoparticles for Targeted Cancer Therapy." in the corresponding sections.
Comments 9
Section 4 1 is extremely long and wordy, Needs breaking up and possibly making moreconcise. Line 323 for example uses the phrase "nano-drug delivery platform" twice and could be clearer
Response 9
Thank you very much for your valuable suggestions! We have made the necessary revisions to the repetitive parts. Regarding the issue that the content of Section 4.1 is a bit lengthy, it is mainly because this section involves a considerable amount of research on macrophage membranes. We have made every effort to optimize and adjust the content and layout to make it more concise and clear.
Comments 10
line 378 not sure what you mean by "green channe!"?
Response 10
Thank you very much for your valuable suggestions! According to your advice, we have changed "green channel" to the more precise expression "efficient penetration pathways" (page 24, line 541).
Comments 11
line 516 language "themembranes"
Response 11
Thank you very much for your valuable suggestions! We have adjusted the format according to your advice and corrected "themembranes" to "the membranes".
Question 12
line 550 check that"NONOate" is defined.
Response 12
Thank you very much for your valuable suggestions! We have added the definition "the NO donor diazeniumdiolate (NONOate)" in the text. We hope this adjustment will make the content clearer.
Comments 13
Line 614 tautology again - "can increase solubility... which is beneficial for increasing theisolubility"
Response 13
Thank you very much for your valuable suggestions! After careful consideration, we agree that this part is indeed not specifically related to the immune cell membrane. Therefore, we have made appropriate deletions to this content. Thank you again for your guidance! (page 32, line 812).
Comments 14
I think the general argument for using immunocyte drugs is weak in line 614-615. Surelyembedding in any membrane increases solubiity, immunocyte membranes are not unique inthis respect? Given that this is key in the article title, this needs more comparison and context。
Response 14
Thank you very much for your valuable suggestions! After careful consideration, we agree that this part is indeed not specifically related to the immune cell membrane. Therefore, we have made appropriate deletions to this content. Thank you again for your guidance! (page 32, line 812).
Comments 15
The figure legends lack information, they should explain what is being shown in the figures.The figures themselves are reasonably clear and effective but legends should explain them.
Response 15
Thank you very much for your suggestions. We have improved each figure legend and provided summaries and explanations for each figure, making it more beneficial for readers’ understanding.
Reviewer 2 Report
Comments and Suggestions for Authors
The review article titled “Tumor Treatment By Nano-Photodynamic Agents Embedded in Immune Cell Membrane” by He et al. has been reviewed where the authors presented a systematic review by investigating and analyzing the advancements in this field. It is understood that non-invasive phototherapy, including photodynamic (PDT) and photothermal therapy (PTT), shows promise in tumor immunotherapy but faces limitations. Recent advances in immune cell membrane-encapsulated nano-photosensitizers enhance targeting and immune response. Bioengineered membranes allow co-delivery of immunotherapy and chemotherapy, enabling synergistic treatment. The article is insightful and timely, however, addressing the following comments would further vitalize it.
- In section 1, authors may consider adding a figure to show the number of publications that have taken place in the recent 5 or 10 years and then explain the trend in the research growth in this field. Both the PDT and PTT can be considered separately and then compared.
- Section 2 needs to be enriched with more information. The comments presented in that section are general and therefore supporting those with concrete evidence by referring to supporting data would be beneficial. To present it more clearly, authors may consider adding a figure or table in this section.
- The caption of Figure 1 is too short and is not self-explanatory. Authors should add an explanation about the figure here so that it may become more informative and easily understandable to readers. Same is also recommended for Figures 2 and 3.
- It seems authors have missed to include many recent works. For example, in line no. 289, the authors have only cited one work for the UCNP related studies. More works and in different areas are to be reviewed and included.
- For section 7, “Perspectives” could be a better word than ‘Discussion’ to present their thoughts and future direction in this section. It would be easier for the readers to understand this section if the authors present their expert opinion and summary through a figure in this section.
- Line No. 687: the chemical formula for hydrogen peroxide to be corrected.
Author Response
Thank you for your valuable comments and suggestions on our manuscript entitled "Tumor Treatment By Nano-Photodynamic Agents Embedded in Immune Cell Membrane" (ID: pharmaceutics-3516044).
We have carefully revised the manuscript according to the reviewers' feedback. Below, we provide a point-by-point response to all comments. All modifications in the revised manuscript are highlighted in Bold font.We hope these revisions address the concerns raised and improve the quality of the manuscript. Please do not hesitate to contact us if further clarification is needed.
Comments 1
In section 1, authors may consider adding a figure to show the number of publications thathave taken place in the recent 5 or 10 years and then explain the trend in the research growthin this field. Both the PDT and PTT can be considered separately and then compared.
Response 1
Thank you for your valuable suggestions! We have added a chart (Figure 1) in the first section, showing the number of papers published in the past ten years that we included and their trends, and have interpreted this trend (page 6, line 165). Since most of the articles we focused on in these studies covered both PDT and PTT simultaneously, we did not discuss them separately.
Comments 2
Section 2 needs to be enriched with more information. The comments presented in thatsection are general and therefore supporting those with concrete evidence by referring tosupporting data would be beneficial. To present it more clearly, authors may consider addinga figure or table in this section.
Response 2
Thank you very much for your valuable suggestions! In response to your advice, we have supplemented relevant supporting data in Section 2 (line 190 on page 7 of the revised text) and cited the corresponding references. At the same time, we also attach great importance to your suggestion on enhancing the visualization of the paper. After careful consideration, we believe that this section has systematically presented the characteristics and existing problems of various nano photosensitizers through a hierarchical discussion. Due to the differences in experimental conditions among different research systems, tabular comparisons may bring certain standardization challenges. Moreover, this section is not the core display part of this paper. To avoid content repetition and excessive complexity, we sincerely ask for your permission to retain the current structure. Thank you again for your patient guidance and support!
Comments 3
The caption of Figure 1 is too short and is not self-explanatory. Authors should add anexplanation about the figure here so that it may become more informative and easily understandable to readers. Same is also recommended for Figures 2 and 3.
Response 3
Thank you very much for your suggestions. We have improved each figure legend and provided summaries and explanations for each figure, making it more beneficial for readers’ understanding.
Comments 4
lt seems authors have missed to include many recent works. For example, in line no. 289, theauthors have only cited one work for the UCNP related studies. More works and in diferentareas are to be reviewed and included.
Response 4
Thank you very much for your suggestions and reminders! Based on your hints, we retrieved 65 relevant articles through the keyword combination of "cell membrane AND UCNP", and found a new article that is very suitable for this review: "Macrophage-reprogramming upconverting nanoparticles for enhanced TAM-mediated antitumor therapy of hypoxic breast cancer". So far, we have not found any articles on UCNP that meet our requirements through other search methods. We have added the relevant content of this article to the text (page 25, line 628). Thank you again for your guidance!
Comments 5
For section 7, "Perspectives" could be a better word than Discussion’ to present theilthoughts and future direction in this section. lt would be easier for the readers to understandthis section if the authors present their expert opinion and summary through a figure in thissection.
Response 5
Thank you very much for your valuable suggestions! According to your comments, we have adjusted the title of this section to "Perspectives" and added Table 4 in the "Perspectives" section to clearly present our summary and viewpoints. Additionally, we have supplemented the summary content about immune cell membranes at line 907 on page 37 and line 918 on page 37, further improving the relevant discussion. Thank you again for your guidance!
Comments 6
Line No.687: the chemical formula for hydrogen peroxide to be corrected.
Response 6
Thank you very much for your comment! We have already corrected the chemical formula of hydrogen peroxide to "H₂O₂". Thank you again for your careful reminder!
Reviewer 3 Report
Comments and Suggestions for Authors
Dear Authors,
Let me congratulate you on such an extensive and complete review paper. It is very well structured and the phenomena are well described. Maybe some complementary explanations about how photodynamic Therapy acts would be interesting to be added. Please, consider the option.
In some parts of the text, more explanations would be necessary for readers to understand better.
Page 2, Lines 64 and 65 require some revision, and on the same page but lines 80 to 82, DC-vaccine-based cancer immunotherapy would require a little more detail.
The case of Tables 1, 2, and 3 is more complex. With the present format, it is very difficult to read. Please, consider reducing the size of the text and adding the lines of the table. It would be interesting to gain visibility and clarity
Many thanks
Author Response
Thank you for your valuable comments and suggestions on our manuscript entitled "Tumor Treatment By Nano-Photodynamic Agents Embedded in Immune Cell Membrane" (ID: pharmaceutics-3516044).
We have carefully revised the manuscript according to the reviewers' feedback. Below, we provide a point-by-point response to all comments. All modifications in the revised manuscript are highlighted in Bold font.We hope these revisions address the concerns raised and improve the quality of the manuscript. Please do not hesitate to contact us if further clarification is needed.
Comments 1
In some parts of the text, more explanations would be necessary for readers to understandbetter.
Response 1
Thank you for your valuable suggestions. We have added some introductions and explanations to certain parts of the text, hoping to make it more convenient for readers to read.
Comments 2
Page 2, Lines 64 and 65 require some revision, and on the same page but lines 80 to 82, DC-vaccine-based cancer immunotherapy would require a little more detail.
Response 2
Thank you for your valuable suggestions! We have revised the extra spaces between "combined treatment" and "be to" on line 64 and 65 of page 2. Additionally, we have supplemented the relevant introduction about cancer immunotherapy based on DC vaccines at lines 80 to 82 of the original text (now line 120 of page 5).
Comments 3
The case of Tables 1. 2, and 3 is more complex. With the present format. it is very difficult toread. Please, consider reducing the size of the text and adding the lines of the table. lt would beinteresting to gain visibility and clarity
Response 3
Thank you for your valuable suggestions! Regarding the issue with the table, our original table is as you have seen, while the one presented in the article is the version modified by the magazine editor, which is indeed beyond our control. We hope you can understand.The journal’s later typesetting should be able to resolve this issue.
Reviewer 4 Report
Comments and Suggestions for Authors
In the review paper "Tumor Treatment By Nano-photodynamic Agents Embedded In Immune Cell Membrane", He and his co-authors described and highlighted several types of membrane carriers for enhanced delivery of photosensitizing agents in PDT. In particular, they focused on immune cell membrane carriers. Although the authors thoroughly reviewed dozens of studies and summarized their results, the review should be improved before publishing as some issues must be addressed:
- According to line 28, this is a systematic review. Thus, a proper section describing the methodology (limitations, inclusion/exclusion criteria) should be implemented.
- I disagree that PDT is regarded as a novel kind of adjuvant therapy (line 38), as it was introduced by Thomas Dougherty in 1975. Please add some historical context to the Introduction section regarding types and generations of photosensitizers and PDT's brief history.
- The authors described pharmacokinetic parameters like permeability and tumor specificity in the main text. However, according to the Introduction, this review focused on studies on encapsulation of photosensitizers to membrane carriers. Therefore, the influence of these carriers on PDT-related parameters, such as singlet oxygen generation or general cytotoxicity, should also be described.
- All the tables provided contain parameters like average particle size, zeta potential, and PDI, which are not discussed within the text. What was the purpose of their showcasing? The authors should at least compare the influence of membrane structure on zeta potential, as it can be seen that this parameter varies in different studies.
- In the Discussion section, I recommend adding a figure or scheme summarizing the findings related to each membrane carrier (macrophage, neutrophil, dendritic cell, etc.).
- Moreover, in the Discussion, the authors should elaborate more on which investigated membrane carriers are the most suitable for photosensitizers in light of reviewed studies.
Author Response
Thank you for your valuable comments and suggestions on our manuscript entitled "Tumor Treatment By Nano-Photodynamic Agents Embedded in Immune Cell Membrane" (ID: pharmaceutics-3516044). We have carefully revised the manuscript according to the reviewers' feedback. Below, we provide a point-by-point response to all comments.
All modifications in the revised manuscript are highlighted in Bold font.We hope these revisions address the concerns raised and improve the quality of the manuscript. Please do not hesitate to contact us if further clarification is needed.
Comments 1
“According to line 28, this is a systematic review. Thus, a proper section describing themethodology (limitations,inclusion/exclusion criteria) should be implemented.”
Response 1
We are grateful to the reviewers for their insightful comments on the classification of the review section. After a careful review, we acknowledge that the original description was inaccurate in terms of the type of review. In response to Question 1, we have revised the relevant part (Page 2, lines 31) to clarify that this study is a narrative review rather than a systematic review.
The corresponding content of the main text is as follows:“This article summarizes and analyzes current research based on the aforementioned advancements.”
Comments 2
“I disagree that PDT is regarded as a novel kind of adjuvant therapy (line 38), as it was introduced by Thomas Dougherty in 1975.Please add some historical context to theIntroduction section regarding types and generations of photosensitizers and PDT's brief history.”
Response 2
Thank you for this important suggestion. We have revised the Introduction section to include:The introduction section incorporates some historical background on PDT, presenting the types and generations of photosensitizers as well as a brief history of PDT.((Page 2, Lines 46; Page 3, Lines57)
Comments 3
“ The authors described pharmacokinetic parameters like permeability and tumor specificity inthe main text. However, according to the Introduction, this review focused on studies onencapsulation of photosensitizers to membrane carriers. Therefore, the influence of these carriers on PDT-related parameters, such as singlet oxygen generation or general cytotoxicityshould also be described”
Response 3
We sincerely appreciate the reviewers' valuable suggestions regarding the biological characteristics of immune cell membrane-coated nanomedicines. As the studies cited in Section 3 (e.g., Hu et al., 2020; Chen et al., 2021) indicate, the reactive oxygen species generation patterns and overall cytotoxicity profiles of different membrane-engineered formulations do indeed show significant consistency. In response to Comment 3, we have provided additional explanations in Section 6 (page 29,line 727).
Comments 4
“All the tables provided contain parameters like average particle size, zeta potential. and PDl which are not discussed within the text. What was the purpose of their showcasing? Theauthors should at least compare the influence of membrane structure on zeta potential, as itcan be seen that this parameter varies in different studies”
Response 4
We sincerely appreciate the reviewer's insightful suggestions regarding the characterization of membrane-coated nanomedicines. In accordance with these suggestions, we have expanded the analysis of the impact of membrane structure on ζ potential in Section 3 (page 12, line 330). However, we would like to clarify that in this specific research context, our systematic review of the cited studies confirmed that the current literature mainly uses particle size and ζ potential as quality control indicators for the completion of membrane coating, rather than as stability indicators. Given this methodological gap, we have added corresponding statements in the discussion section (page 40, line 956). This addition is consistent with the reviewer's concerns and points out a key direction for methodological improvement in this field.
Comments 5
“In the Discussion section,l recommend adding a figure or scheme summarizing the finding related to each membrane carrier (macrophage, neutrophil, dendritic cell. etc.).”
Response 5
We sincerely appreciate the reviewers' constructive suggestions regarding a systematic comparison of the immune cell membrane-based delivery systems. In response to this comment, we have included a comprehensive comparative analysis in the discussion section (Section 8, page 37), namely Table 4 ("Comparative Characteristics of Immune Cell Membrane-Based Drug Delivery Systems").
Comments 6
“Moreover, in the Discussion, the authors should elaborate more on which investigated membrane carriers are the most suitable for pohotosensitizers in that of reviewed studies”
Response 6
We sincerely appreciate the insightful comments from the reviewers regarding the optimization of photosensitizer carriers. In accordance with this suggestion, we have expanded Section 8 (page 36, line 907) with the following new content:”We believe that, based on current research, macrophage membranes are considered the most suitable carriers for photosensitizers among immune cell membranes due to their natural tumor chemotactic ability, ROS generation induced by M1 polarization, and synergistic immune regulatory effects. Macrophage membranes have demonstrated highly efficient synergy in PDT/chemotherapy/immunotherapy in models such as glioma and breast cancer, achieving a relatively high level of technological maturity. T cell membranes have irreplaceable advantages in the treatment of brain tumors, while NK cell membranes show significant potential in controlling metastatic foci, and can be used as supplementary options in specific scenarios. In the future, their adaptability needs to be further enhanced through engineering modifications (such as targeted modification of membrane proteins and hybrid membrane design).”
Round 2
Reviewer 1 Report
Comments and Suggestions for Authors
I think the changes address my concerns.
Author Response
Thank you for your recognition!
Reviewer 2 Report
Comments and Suggestions for Authors
The authors have replied to the comments satisfactorily.
Author Response
Thank you for your recognition!
Reviewer 4 Report
Comments and Suggestions for Authors
Dear Authors,
All my comments have been addressed. I haven't noticed any further issues thus the paper can be accepted in its present form.
Author Response
Thank you for your recognition!